**Data Availability Statement:** All relevant data are within the manuscript and its Supporting information files.

**Funding:** This study was supported by The National Natural Science Foundation of China

# Study on the control effect of local basement replacement on the stability of dump

**Dong Wang, Li Yin** **\*, Xiaoyu Xing**

College of Mining, Liaoning Technical University, Fuxin, China

\* azhuang_001@163.com

## Abstract

This study focuses on effectively controlling landslides at the boundary of a soft rock open-pit dump while ensuring safe increases in the dump's capacity and optimal utilization of external dump sites. To achieve this, the adoption of a local filling method for the dump base is proposed. By leveraging the concepts of limit equilibrium theory and equivalent shear strength parameters, the mathematical expression of the slope stability coefficient in the Morgenstern-Price method is derived and improved. This improved method is then applied to a real engineering example to determine the optimal basement replacement rate required to maintain slope stability. The findings reveal that the local filling of the base is well-suited for slopes susceptible to potential landslides associated with cutting layers, bedding layers, and swelling. For practical ease, it is advisable to choose the lowest step in the dump's slope for construction convenience. As the local replacement rate of the base increases, the slope's stability coefficient gradually improves, with the K-Fs ratio showing a prominent role in this process. Additionally, numerical simulation methods are employed to elucidate the mechanism of the dump's landslide following local basement replacement, thereby providing comprehensive evidence of the engineering applicability of this method. The research results demonstrate a promising practical application prospect for effectively controlling the stability of soft base dump slopes.

## Introduction

The stability of slopes is fundamental to ensuring the efficiency, safety, and environmental sustainability of open-pit mines. It directly impacts the well-being of engineering personnel, equipment integrity, and the protection of nearby facilities. However, the intricate geological conditions prevalent in mines, such as weak basement layers [1–3], groundwater influence [4, 5], and basement inclination [6], often impose limitations on slope stability, making slopes prone to instability. Consequently, the formulation of cost-effective and rational measures for slope stability control becomes paramount. In the pursuit of optimizing land resource utilization, numerous external dumps extend from the boundaries of the acquired land. Yet, in the event of significant deformation or, worse, a landslide within these slopes, conventional stability control approaches can inadvertently lead to issues of unlawful land utilization and related legal complexities. Hence, the exploration of slope stability control methods gains exceptional

(52204135, 52204136), Liaoning Tens of Millions of Talents Program, Silk Road 1+1 Scientific Research Cooperation Project of Ministry of Education(P20210121076), Discipline Innovation Team of Liaoning Technical University (LNTU20TD-01) and International Science and Technology Cooperation Program of Liaoning Province(2022JH2/10700004). The funders had no role in study design, data collection and analysis, decision to publish, or preparation of the manuscript.

**Competing interests:** The authors declare that they have no conflicts of interest.

significance, not only for addressing immediate concerns but also for potential wider application in the stability management of analogous open-pit dump settings.

In the context of dump slopes with soft foundations, substantial research efforts have been dedicated to understanding and mitigating instability. Numerical simulations have been utilized to analyze key factors and instability patterns in dump slopes, offering insights into the underlying mechanisms [7]. Investigations combining model tests, numerical simulations, and theoretical deductions have elucidated failure modes in dump slopes with weak basements, emphasizing the role of weak basement positioning in influencing slope failures [8, 9]. Quantitative assessments of slope stability, particularly in specific regions, have employed two-dimensional finite element numerical simulations to determine optimal design heights for varying dumping scenarios [10]. Scholars have proposed diverse measures to enhance dump slope stability, including basement treatments, internal waterproofing, and manipulation of discharge parameters [11]. Furthermore, comprehensive studies on slope treatment in the Heidaigou Coal mine dump have yielded insights into construction techniques, runoff management, and vegetation allocation [12]. Innovative strategies, such as "cutting the top, strengthening, and pressing the foot," have been identified for effectively addressing dump slope accumulation [13]. Comparative assessments employing limit equilibrium methods have underscored the efficacy of foot pressing measures, particularly for surfaces prone to sliding at the lower slope sections [3]. Scholarly contributions have proposed holistic approaches for landslide-prone areas, encompassing measures like clearance, load reduction, anti-slide pile retention, and integrated water management, collectively bolstering slope stability [14].

The selection of slope stability control strategies involves careful consideration of multiple factors, encompassing technical feasibility, construction circumstances, environmental consequences, and economic viability. Despite effective control methodologies having been proposed by scholars and professionals under specific engineering geological conditions, the prevalence of soft soil basements remains a notable challenge in the Xilin Gol League region of China. To illustrate, in the context of the Guole No. 2 open-pit coal mine in Jilin, a distinctive slope angle of 10° characterizes a landslide-prone deformation zone within the dump. This unique situation significantly impacts subsequent soil discharge projects. In such cases, the local filling approach for the dump base emerges as a particularly suitable solution. Leveraging the concept of equivalence, this study deduces the interrelation of equivalent shear strength parameters within the localized basement filling area, thereby refining the expression of stability coefficients within the framework of the Morgenstern-Price method. This theoretical advancement provides a foundation for applying the local filling method in addressing dump slopes with soft basements.

## Study on the method of local replacement of dump base

### Design of base partial replacement scheme

The technique of local basement replacement serves as a prevalent method in civil engineering, primarily employed to fortify soft foundation soil. Often, the quaternary layer underlying dump sites consists of sandy soil, characterized by high water permeability and a structurally weak interface with the underlying Tertiary clay. The resulting feeble base layer significantly contributes to the instability of dump slopes. By adopting the local filling approach at the base, it becomes possible to modify the shear strength parameters of the weak layer through a combination of the filling material and the original foundation. In essence, this method alters the mechanical properties of the weak layer, thereby exerting control over the stability of the dump slope.

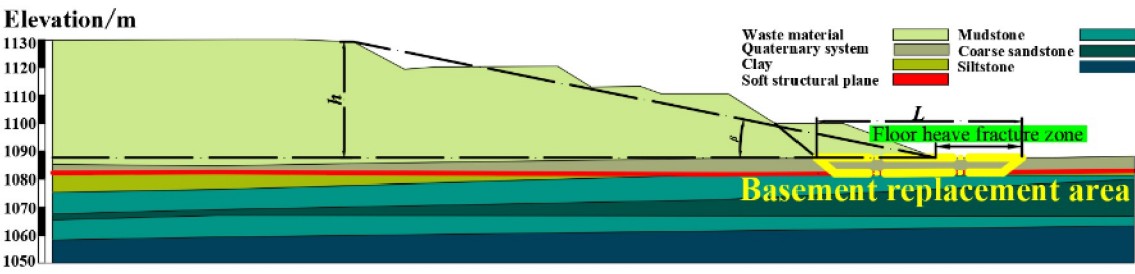

**Fig 1. Schematic diagram of the current slope basement replacement area.**

Given that the open pit mine's base is concealed and complete material removal isn't practical, a viable strategy involves siting the basement replacement within the scope of the dump's lowest step (refer to Fig 1). This approach, as depicted in Fig 1, stands as a highly feasible engineering solution. By stripping the bottom step of the dump and the quaternary layer of the basement, the weak interlayer structural plane is exposed and disrupted. Ultimately, the void is replenished with robust rock and soil materials possessing enhanced mechanical properties.

## Derivation of local filling basis theory

Strengthening the weak layer beneath the dump base enhances the overall rock and soil mass, effectively treating it as a homogeneous material. Its shear strength approximates that of natural clay mixed with the weak layer in a specific proportion. To quantify this process, the concept of the local filling rate 'K' is introduced. K represents the ratio of the base's filled section to the cover section 'L' of the subsequent dump step. The formula for K is defined as $K = a/L$. The actual filling involves uniformly and equidistantly depositing natural clay material within the base's filling area. Presently, a well-established analytical method for determining the shear strength parameters of this newly formed composite weak layer is lacking.

Varying filling rates 'K' yield distinct shear strength parameters for the composite weak layer. During shearing along the potential sliding surface of a slope where the combined weak layer acts as the base interface, both the unfilled section of the base and the replaced rock and soil segment experience shear forces. The composite weak layer within the replacement area is treated as an integrated unit, characterized by equivalent cohesion 'c'' and equivalent internal friction angle '$\varphi$''. When employing the Mohr-Coulomb criterion to analyze slope stability, it becomes imperative to consider the shear strength parameters at different levels within the rock and soil mass (such as the natural soil layer and weak basement layer). Combining these parameters enables the calculation of equivalent shear strength parameters. Specifically, the equivalent cohesion denotes the cohesion linked to the equivalent shear strength parameter. The Mohr-Coulomb criterion is mathematically expressed as follows:

$$\tau_f = c + \sigma_n \tan \varphi \tag{1}$$

Where $\tau_f$ is the shear stress on the sliding surface; c and $\varphi$ are the cohesion and internal friction angles of each rock layer passed by the sliding surface, respectively. σn is the normal stress of the strip on the sliding surface.

When the base is locally replaced, the rock and soil mass can be regarded as a heterogeneous material. The shear strength parameters of the combined weak layer are set as c' and $\varphi$', and the expressions of the equivalent cohesive force and equivalent internal friction Angle of

the combined weak layer are as follows:

$$c' = K_1 c_1 + (1 - K_1)c_0 + \Delta c \tag{2}$$

$$\varphi' = \arctan(K_2 \tan \varphi_1 + (1 - K_2) \tan \varphi_0 + \Delta \tan \varphi) \tag{3}$$

Among them, c1 is the cohesion of natural clay, $c_0$ is the cohesion of weak layer, $K_1$ and $K_2$ are the replacement rate of cohesion and internal friction Angle respectively, and $\Delta c$ represents the influence of local replacement. In general, $\Delta c$ is a position-dependent function, and $\varphi_1$ is the internal friction Angle of natural clay. $\varphi_0$ is the friction Angle in the weak layer. $\Delta \tan \varphi$ represents the equivalent increment of internal friction Angle caused by local filling.

$$\Delta c = f(x, y) \quad \Delta \tan \varphi = g(x, y) \tag{4}$$

Where $x$ and $y$ represent the horizontal and vertical coordinates of the weak layer respectively. Therefore, the equivalent cohesion $c'$ and the equivalent internal friction Angle $\varphi'$ can also be regarded as a position-dependent function. When the weak layer occurs nearly horizontally, it can be assumed that the shear strength parameters of the weak layer change smoothly at different positions, namely $\Delta c = 0$, $\Delta \tan \varphi = 0$.

The Morgenstern-Price method, which is widely used in two-dimensional rigid body limit equilibrium method [15–17], was adopted to improve the formula, analyze the equation of force balance and moment balance of the slide-block of slope, evaluate the stability of slope, and consider the equivalent cohesion and internal friction Angle of the weak layer in the local displacement area of the base, as well as the original weak layer in the undisplacement area. The theoretical derivation formula of slope stability with the combination of two weak layers.

The height of the slope is $H$, the slope is $\beta$, the width of the local filling area of the basement is $L$, the thickness of the weak layer is $h_1$, the thickness of the weak layer in the unfilled area is $h_0$, and the stability coefficient of the slope is $Fs$. The slope is divided into two parts, that is, the area of basement partial filling and the area of unfilled filling. For the local filling area of the base, the equivalent shear strength is $c'$ and the equivalent internal friction Angle is $\varphi'$; for the unfilled area, the shear strength is $c_0$ and the internal friction Angle is $\varphi_0$ (Fig 2).

For the weak layers in the locally filled and unfilled regions of the basement, their dead weight and earth pressure are considered respectively, that is, their effective stresses $\sigma_{v1}$ and $\sigma_{v0}$

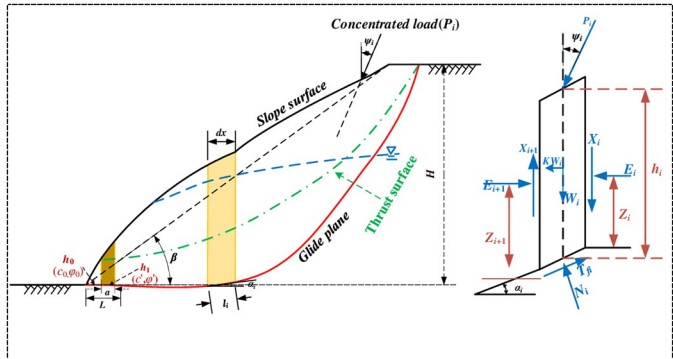

**Fig 2. Improved morgenstern-price mana model.**

are calculated respectively. For the local replacement area of the base, there are:

$$\sigma_{v1} = \gamma_1 h_1 + (\gamma_1 - \gamma_w)L \tag{5}$$

Where $\gamma_1$ is the weight of the replacement fill and $\gamma_w$ is the weight of water.

For the unfilled area, there are:

$$\sigma_{v0} = \gamma_0 h_0 \tag{6}$$

Where $\gamma_0$ is the severity of the original weak layer.

Assuming that the length of a section on the slope is $L_1$, the dead weight of the strip per unit thickness is w, and the earth pressure is $p$, then the moment $M$ of the unit is:

$$M = (w - p)L_1{}^2/2 \tag{7}$$

The slope is divided into several strips, and the torque of each unit is $M_1$, $M_2$..., $M_n$. The filling rate $K_i$ and $K'_i$ of shear strength and internal friction Angle were calculated respectively for the local filling area and unfilled area of the base, as follows:

$$K_i = \frac{(c' + \sigma_{v1} tan\varphi')}{\sigma_{v1}\, tan(\beta - \varphi')} \tag{8}$$

$$K'_i = \frac{c_0}{\sigma_{v0}\, tan(\beta - \varphi_0)} \tag{9}$$

For each bar, calculate its stability coefficient $F_i$, as follows:

$$F_i = \frac{2\sum M_i + K_i \sigma_{v1} Lh_1{}^2}{Ki'\sigma_{v0}Lh_0{}^2 + \gamma wH sin\beta L_1^{2}} \tag{10}$$

The sigma $M_i$ is the sum of the torques of all the bars on the slope.

For the whole slope, calculate its average stability coefficient $F_{av}$, as follows:

$$F_{av} = \frac{\Sigma F_i}{\Sigma W_i} \tag{11}$$

Where $F_i$ is the stability coefficient of the weak layer in section I, and $W_i$ is the gravitational action of the weak layer in section I.

Specifically, $F_i$ can be calculated by the formula (10), while $W_i$ can be calculated by the following formula:

$$W_i = \gamma_i H_i S_i \tag{12}$$

Where $\gamma_i$ is the unit weight of the weak layer in the I section, $H_i$ is the thickness of the weak layer in the I section, and $S_i$ is the area of the weak layer in the slope.

According to formulas (8) and (9), the equivalent strength parameter and internal friction Angle of the weak layer in each segment can be obtained, which are $c'_i$ and $\varphi'_i$ respectively. By multiplying them by the area Si of the corresponding segment, the strength $S'_i$ and shear strength $S'_i tan(\varphi'_i)$ of the weak layer in this segment can be obtained, and then they are added respectively. The equivalent strength parameter $c'$ and internal friction Angle $\varphi'$ of the whole

slope are obtained:

$$c' = \frac{\Sigma S_i' c_i'}{\Sigma S_i'} \tag{13}$$

$$tan\varphi' = \frac{\Sigma Si' tan(\varphi i')}{\Sigma S_i'} \tag{14}$$

Where, $S_i' = S_i cos^2(\varphi_i' - \beta) + \frac{S_i sin^2(\varphi_i' - \beta)}{N_q + N_\gamma tan^2(\varphi_i' - \beta)}$

The calculated average equivalent strength parameter and internal friction Angle are put into the formula of the Morgenstern-Price method, then the stability coefficient Fs of slope can be obtained:

$$Fs = \frac{c'N_q + Htan\varphi'N_c + 0.5\gamma H^2 tan\varphi'N_\gamma}{W} \tag{15}$$

Among them, $N_q$, $N_c$, $N_\gamma$, and $W$ are the coefficients in the Morgenstern-Price method, which can be calculated according to the slope geometry and soil parameters.

## Engineering example

The waste dump of Jilin Guole No. 2 open-pit mine in the Inner Mongolia Autonomous Region of China is located on the south side of the stope with a horizontal base. Seasonal river water and atmospheric precipitation infiltrate the underground, resulting in the distribution of a Quaternary pore aquifer throughout the region. This aquifer is primarily composed of medium-fine sand and gravel. Underneath the Quaternary system lies a substantial red clay layer of the Tertiary system, which acts as a robust water barrier. The dump's southern side extends to the boundary and has given rise to the creation of four discharge steps. Each step measures 10 meters in height, with a platform width of 20 meters and a slope angle of 33˚. The overall slope height is 40 meters, and the initial slope angle is 10˚. The last step's slope angle has been modified to 22˚. Notably, transverse cracks are prevalent at distances ranging from 44 to 83 meters from the +1130 level's top line. Within 30 meters from the slope's base, floor heave phenomena have emerged. An illustration of the dump's slope engineering is provided in Fig 3.

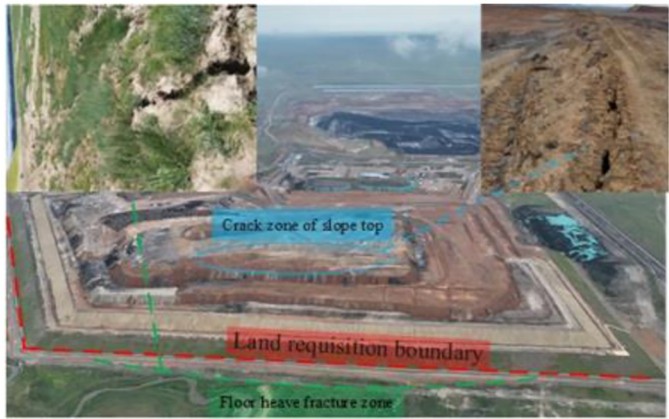

**Fig 3. DEM map of the deformation area of dump.**

**Table 1. Selected values of physical and mechanical indexes of rock and soil mass.**

| Geotechnical designation | Density $\gamma$(kN/m3) | Cohesion $c$(kPa) | Friction $\varphi$(°) | Modulus of elasticity $E$(Mpa) | Poisson $\mu$ |
| --- | --- | --- | --- | --- | --- |
| Quaternary system | 18.0 | 5.0 | 23.9 | 1.3 | 0.18 |
| Clay (natural) | 19.8 | 65.8 | 19.6 | 7.65 | 0.25 |

Based on thorough field investigations into the dump site's slope deformation, a discernible weak structural plane has formed at the contact layer between the Quaternary water-rich sand gravel layer and the underlying Tertiary clay layer at the outer dump site's base. Consequently, a potential sliding body has taken shape. To effectively ensure slope stability while accommodating the dump's subsequent expansion, the engineering approach of partially filling the basement has been employed as guidance.

Compliance with relevant specifications [18, 19] led to the selection of a safety reserve factor of 1.1 for the slope in the deformation area of the dump. This choice takes into account factors such as the slope's service life, its significance, the understanding of rock and soil mass indices, the potential impact of landslides, and future dump expansion requirements.

### The coupled soil parameters in the replacement area are determined research

The physical and mechanical parameters of the quaternary sand in the base of the dump site and the clay in the natural mining area are tested by the geotechnical laboratory, and the physical and mechanical parameters of the base rock and soil are obtained (Table 1).

The soil mass is subject to the load of the dump in the basement replacement project. Considering the mutual coupling of the two soil materials, and the extraction of the soil sample after coupling is relatively difficult, the numerical simulation of the coupled soil mass based on the finite element numerical method and combined with the different physical and mechanical parameters of the two materials can predict the mechanical characteristics of the coupled soil mass under different working conditions. FLAC3D establishes a three-dimensional geometric model, including the boundary and interface of the foundation pit, respectively fills the soil of two Mohr-coulomb materials, and defines the initial boundary conditions and loading modes (Fig 4). The process of loading on the surface of the foundation pit was simulated, and parameters such as time, displacement and stress of

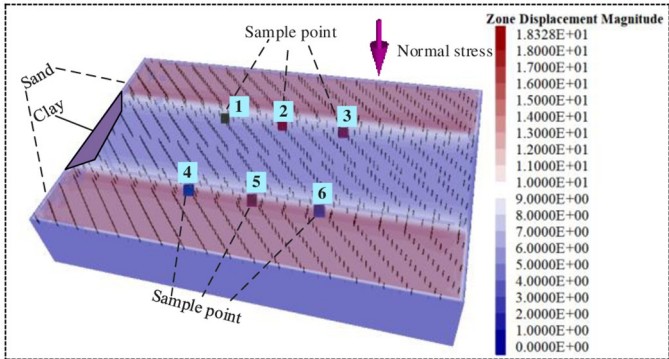

**Fig 4. Numerical simulation test of coupled soil.**

sample points in the coupled soil were recorded. The data in the loading process were analyzed and processed to obtain the stress-strain relationship curve of the coupled soil (Fig 5). The shear strength and denaturation parameters of coupled soil were calculated (Table 2).

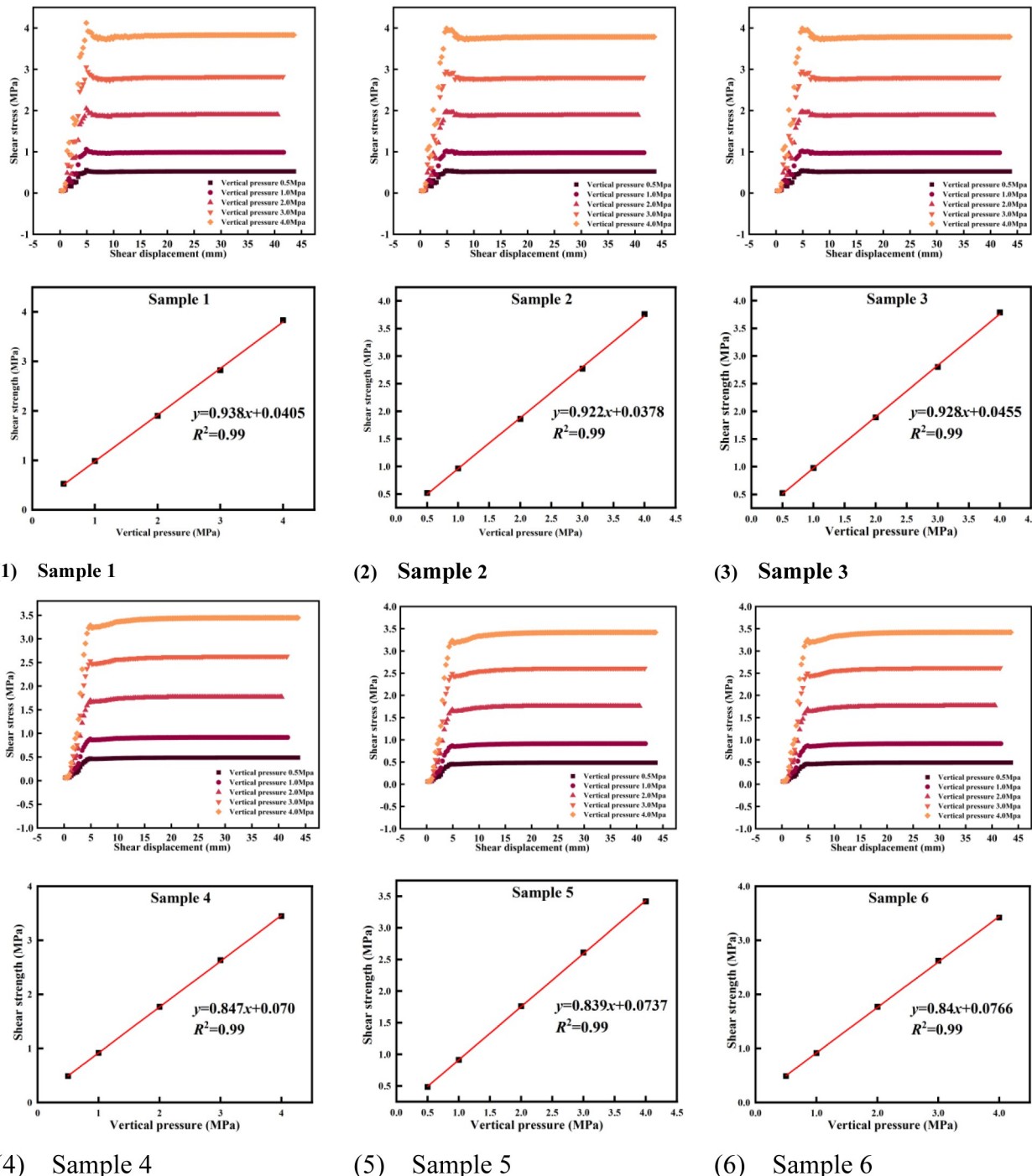

**Fig 5. Coupled soil mechanical parameter calculation diagram stress-strain relationship at sample point, τ-σ elationship diagram calculation.**

**Table 2. Shear strength parameters and denaturation parameters of coupled soil.**

| Geotechnical designation | Density $\gamma$(kN/m3) | Cohesion $c$(kPa) | Friction $\varphi$(°) | Modulus of elasticity $E$(Mpa) | Poisson $\mu$ |
| --- | --- | --- | --- | --- | --- |
| Coupled soil | 18.9 | 32.0 | 20.9 | 6.64 | 0.22 |

### Relationship between replacement rate and slope stability alculation

When the local filling rate of the base '$K$' is 1, it means that all the covered area of the next step of the dump is replaced with clay; when the local filling rate of the base $K$ is 0, it means that no treatment has been done in the covered area of the next step of the dump. In the current slope deformation area of the dump site, it is assumed that the bottom step is replaced with the base, and the strength reduction method is used to select the reduction coefficient $R$ and reduce the shear parameters $c$ and $\varphi$ of natural clay at the same time, to obtain the stability coefficient satisfying the safety reserve coefficient. According to Eqs (2) and (3), the shear strength parameters of the weak layer in the filling area are determined comprehensively by the two coefficients of filling rate $K_1$ and $K_2$. Therefore, firstly, the weak layer equivalent cohesion replacement rate $K_1$(i.e., reduction coefficient $R$) is determined, then the replacement rate $K_2$ of the equivalent internal friction Angle $\varphi$ is relatively deduced, and finally, the minimum value of both is selected to determine the final replacement rate $K$, i.e., $K = min(K_1, K_2)$. In this case, the replacement rate K is the exact solution of the equivalent shear strength parameter and the determination process of the replacement rate K (Fig 6).

The calculation result of slope stability corresponding to the minimum replacement rate $K_{min}$ is determined according to the program diagram (Fig 7), The relationship between replacement rate $K$ and slope stability coefficient $Fs$ is also given (Table 3).

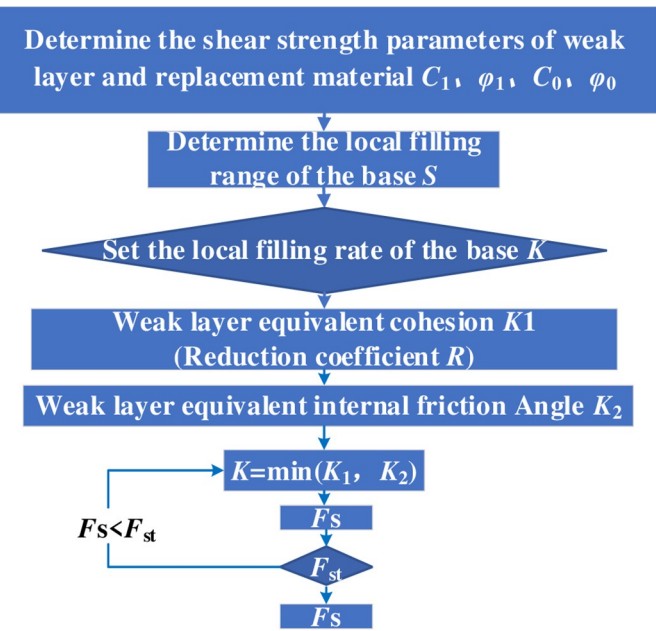

**Fig 6. Block diagram of the program.**

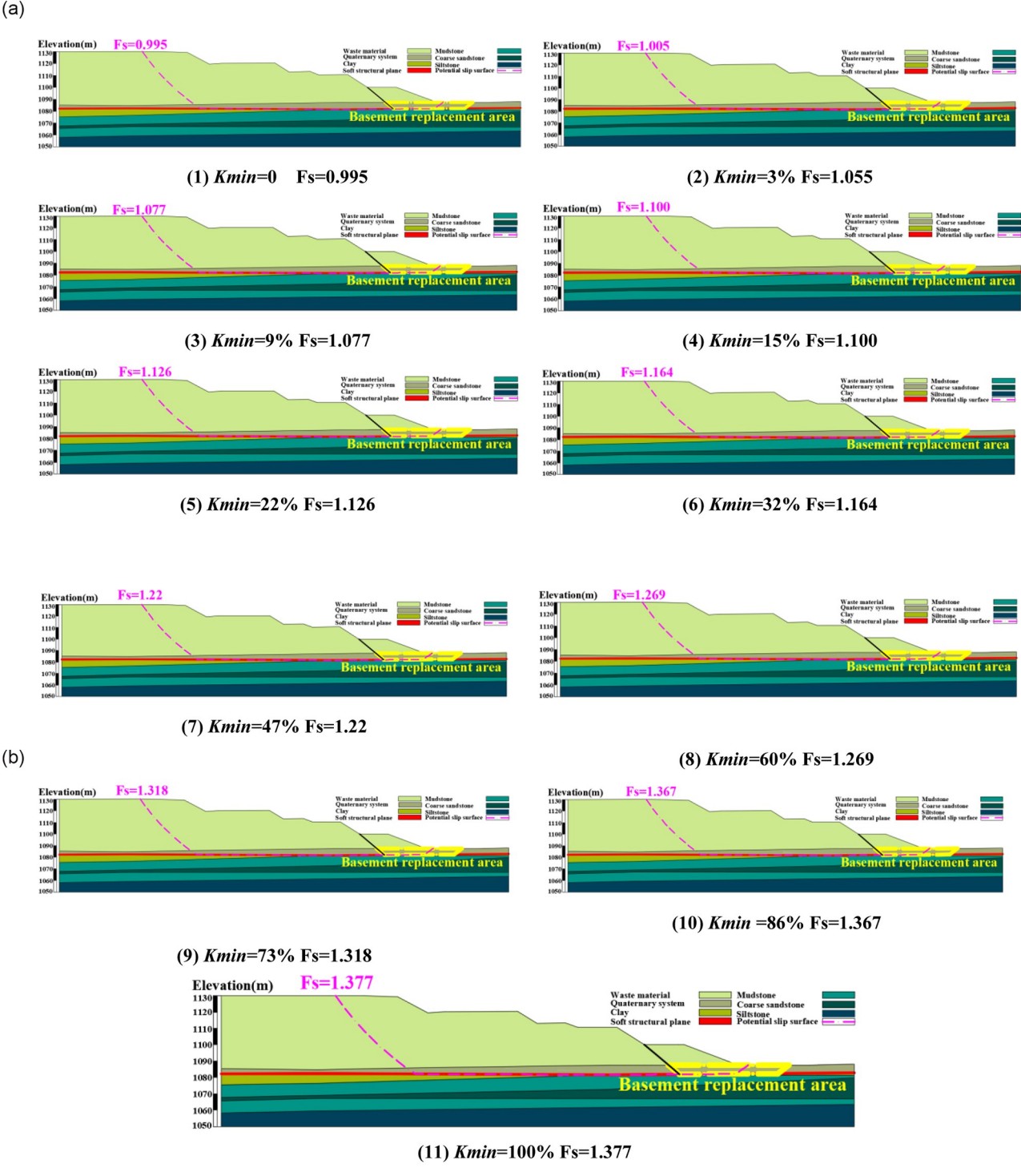

**Fig 7. Slope stability coefficient Fs corresponding to different filling rate K.**

**Table 3. Relationship between *K* and slope stability coefficient *Fs*.**

| Reduction coefficient *R* (%) | Reduce cohesion *c'*((kPa)) | Reduce friction *φ'* (°) | *Fs'* | *Kc*(%) | *Kφ*(%) | Equivalent cohesion *c'*((kPa)) | Equivalent friction *φ'* (°) | *Fs* |
|---|---|---|---|---|---|---|---|---|
| 0 | 0 | 4.4 | 0.995 | 0 | 0 | 0.00 | 4.40 | 0.995 |
| 25 | 16.45 | 4.9 | 1.049 | 25 | 3 | 1.97 | 4.88 | 1.055 |
| 30 | 19.74 | 5.88 | 1.070 | 30 | 9 | 5.92 | 5.83 | 1.077 |
| **35** | **23.03** | **6.86** | **1.120** | **35** | **15** | **9.87** | **6.78** | **1.100** |
| 40 | 26.32 | 7.84 | 1.141 | 40 | 22 | 14.48 | 7.88 | 1.126 |
| 50 | 32.9 | 9.8 | 1.155 | 50 | 32 | 21.06 | 9.44 | 1.164 |
| 60 | 39.48 | 11.76 | 1.198 | 60 | 47 | 30.93 | 11.76 | 1.22 |
| 70 | 46.06 | 13.72 | 1.242 | 70 | 60 | 39.48 | 13.74 | 1.269 |
| 80 | 52.64 | 15.68 | 1.286 | 80 | 73 | 48.03 | 15.68 | 1.318 |
| 90 | 59.22 | 17.64 | 1.33 | 90 | 86 | 56.59 | 17.59 | 1.367 |
| 100 | 65.8 | 19.6 | 1.377 | 100 | 100 | 65.80 | 19.60 | 1.377 |

## Result discussion

The K-Fs scatter plot was drawn and linear fitting was carried out (Fig 8). The expression of the fitting linear equation is

$$Fs = 0.0037K + 1.037 \tag{16}$$

The correlation coefficient $R^2$ is 0.98, with a high degree of fitting, which can accurately describe the relationship between the local filling rate K and the slope stability coefficient *Fs*. It can be seen that with the increase of the local filling rate of the basement, the slope stability coefficient gradually increases, showing an approximate first-order function relationship.

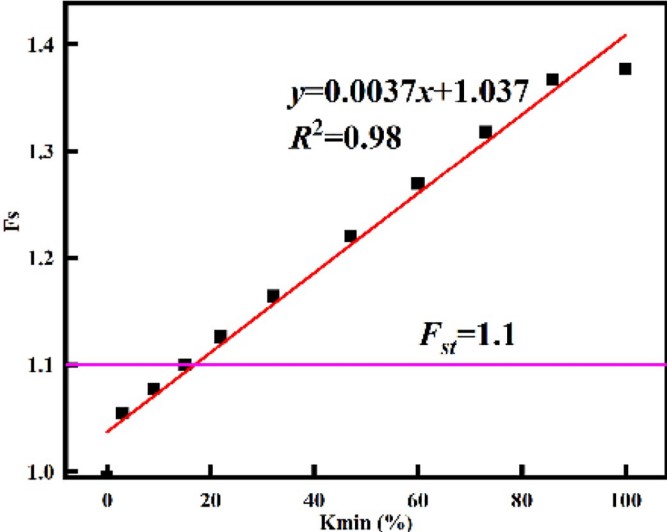

**Fig 8. Relation curve between slope stability coefficient *Fs* and replacement rate *K*.**

## Numerical simulation

To study the influence of the basement local filling method on the landslide mode and mechanism of open-pit mine slope, the author takes the slope in the deformation area as the research object, establishes a three-dimensional numerical simulation model, and carries out the stability study. By analyzing the displacement, stress, deformation, and failure characteristics of the slope, the reliability of the method of partial basement replacement is verified.

**Establishment of model.** According to the aerial photography data, the slope elevation data of the dump were obtained. Combined with the formation elevation data from the geological exploration results, the 3D numerical model was established with Rhino and the grid was divided, including 428996 grids and 79014 nodes. Finally, the model was imported into FLAC3D. The three-dimensional dump slope model in FLAC3D is shown in Fig 9. Regional strata include the bedrock layer, Tertiary clay layer, quaternary sand gravel layer, and discharged materials.

**Numerical simulation results and analysis.**

1. *Displacement distribution characteristics*. Fig 10a shows the three-dimensional and two-dimensional displacement program of the dump slope. It can be seen that in the slope inclination direction, the terrain of the Tertiary clay layer as a water-barrier layer gradually decreases, and the groundwater flows and aggregates to the south, which leads to the deterioration of mechanical properties of the contact surface between the Quaternary system and the Tertiary system, and thus the increase of displacement.

2. *Stress distribution characteristics*. Fig 10b and 10c respectively show the two-dimensional horizontal and vertical stress distribution of the slope. It can be seen that tensile stress zones have been formed on horizontal slopes in the horizontal direction. As the tensile strength of the rock body itself is much lower than that of compressive strength, tensile cracks may be formed on the slope surface, and the potential landslide mode is cut bedding—along bedding—swelling. In addition, some rock and soil bodies have been deformed and damaged. The deformation parameters (elastic modulus and Poisson's ratio) are changed, and the local stress concentration occurs at the fracture site. In the vertical direction, the stress distribution of the slope body is uniform, but there is no obvious irregular change. Under the surface of the slope body, it presents a linear upward trend with the increase of buried

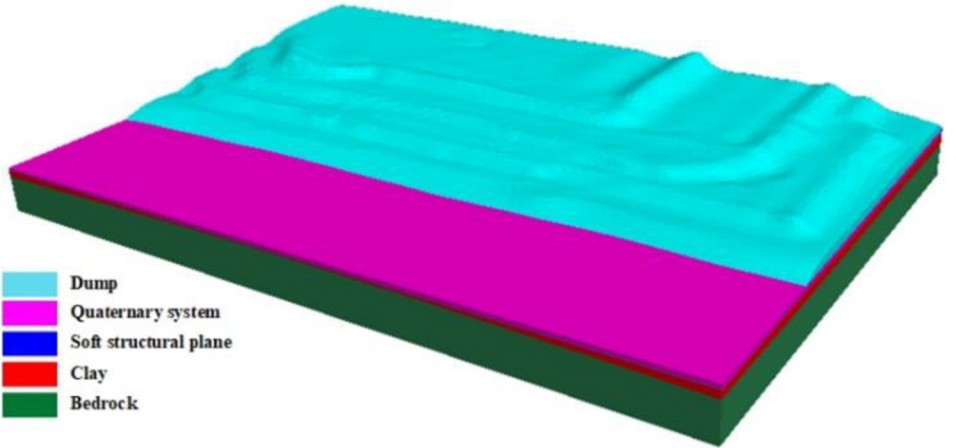

**Fig 9. Numerical model of three-dimensional dump.**

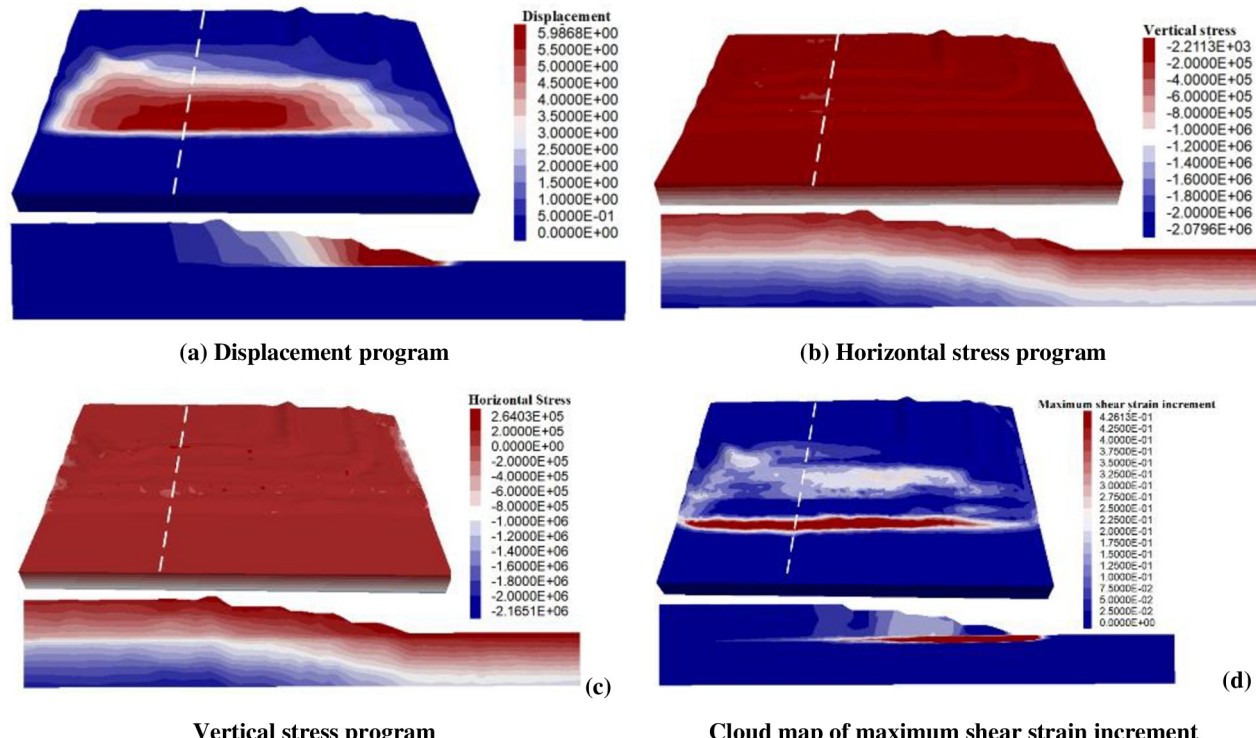

**(a) Displacement program**

**(b) Horizontal stress program**

**(c)**

**Vertical stress program**

**(d)**

**Cloud map of maximum shear strain increment**

**Fig 10. Numerical simulation results.**

depth, indicating that the distribution of vertical stress in the slope body is mainly affected by the gravity weight of rock and soil mass.

3. *Distribution characteristics of shear strain increment*. Fig 10d shows the program of the maximum shear strain increment at the critical instability of the slope. It can be seen that the shear strain increment of the weak layer at the interface between the Quaternary system and the Tertiary system in the basement is significant, and the weak layer of the basement produces plastic yield under the influence of the gravity of the discharged material. Shear failure plays a decisive role in rock mass failure type, and the shear strain increment map can indicate the potential sliding mode of landslide.

## Conclusion

1. The slope stability can be effectively controlled by replacing the next step of the waste dump and changing the mechanical characteristics of the weak structural plane of the basement.

2. Based on the limit equilibrium method and combined with the idea of equivalent shear strength parameters, the theoretical calculation method of slope stability of the local filling base was derived, which laid a theoretical foundation for the study of slope treatment schemes under different working conditions.

3. Combined with the stability results under different filling rates K, the approximate first-order function relationship is obtained. It can be seen that with the increase of the local filling rate of the basement, the stability coefficient of the slope gradually increases.

4. The numerical simulation outcomes unveil a predominant potential landslide mode within the dump's landslide region, characterized as a shear-bedding-bedding-bulging slide with the weak bed serving as the basal interface. In line with these findings, a strategic replacement of the next dump step with natural clay, covering 15% of the area, has been determined. Remarkably, this approach not only fulfills the safety reserve criteria for slope stability but also demonstrates a notably enhanced treatment efficacy. Furthermore, the stability outcomes deduced from numerical simulations exhibit a strong concurrence with those derived from the two-dimensional rigid limit equilibrium method. The alignment encompasses both the morphology and positioning of the potential slip surfaces identified by the two methodologies.

## Supporting information

**S1 Data.**
(XLS)

**S2 Data.**
(XLSX)

**S3 Data.**
(XLS)

## Author Contributions

**Conceptualization:** Li Yin.

**Data curation:** Li Yin, Xiaoyu Xing.

**Funding acquisition:** Dong Wang.

**Investigation:** Li Yin.

**Methodology:** Dong Wang, Li Yin.

**Resources:** Dong Wang.

**Supervision:** Li Yin.

**Writing – original draft:** Li Yin.

**Writing – review & editing:** Li Yin.

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
