## [Decision Letter · Decision Letter 0]

10 Jul 2023

PONE-D-23-16609Study on the control effect of local basement replacement on the stability of dumpPLOS ONE

Dear Dr. Li,

Thank you for submitting your manuscript to PLOS ONE. After careful consideration, we feel that it has merit but does not fully meet PLOS ONE’s publication criteria as it currently stands. Therefore, we invite you to submit a revised version of the manuscript that addresses the points raised during the review process.

We look forward to receiving your revised manuscript.

Kind regards,

Sani Isah Abba, PhD

Academic Editor

PLOS ONE

Journal Requirements:

   "This study was supported by The National Natural Science Foundation of China (52204135, 52204136), Liaoning Tens of Millions of Talents Program, Silk Road 1+1 Scientific Research Cooperation Project of Ministry of Education(P20210121076), Discipline Innovation Team of Liaoning Technical University 

(LNTU20TD-01) and International Science and Technology Cooperation Program of Liaoning Province(2022JH2/10700004)."

   "This study was supported by The National Natural Science Foundation of China (52204135, 52204136), Liaoning Tens of Millions of Talents Program, Silk Road 1+1 Scientific Research Cooperation Project of Ministry of Education(P20210121076), Discipline Innovation Team of Liaoning Technical University (LNTU20TD-01) and International Science and Technology Cooperation Program of Liaoning Province(2022JH2/10700004)."

   "This study was supported by The National Natural Science Foundation of China (52204135, 52204136), Liaoning Tens of Millions of Talents Program, Silk Road 1+1 Scientific Research Cooperation Project of Ministry of Education(P20210121076), Discipline Innovation Team of Liaoning Technical University (LNTU20TD-01) and International Science and Technology Cooperation Program of Liaoning Province(2022JH2/10700004)."

6. We note that Figure 2 in your submission contain [map/satellite] images which may be copyrighted. All PLOS content is published under the Creative Commons Attribution License (CC BY 4.0), which means that the manuscript, images, and Supporting Information files will be freely available online, and any third party is permitted to access, download, copy, distribute, and use these materials in any way, even commercially, with proper attribution. For these reasons, we cannot publish previously copyrighted maps or satellite images created using proprietary data, such as Google software (Google Maps, Street View, and Earth). For more information, see our copyright guidelines: http://journals.plos.org/plosone/s/licenses-and-copyright.

7. Please ensure that you refer to Figures 8 and 9 in your text as, if accepted, production will need this reference to link the reader to the figure.

8. We note you have included a table to which you do not refer in the text of your manuscript. Please ensure that you refer to Table 3 in your text; if accepted, production will need this reference to link the reader to the Table.

Reviewers' comments:

Reviewer's Responses to Questions

**Comments to the Author**

1. Is the manuscript technically sound, and do the data support the conclusions?

Reviewer #1: No

Reviewer #2: Yes

2. Has the statistical analysis been performed appropriately and rigorously? 

Reviewer #1: No

Reviewer #2: N/A

3. Have the authors made all data underlying the findings in their manuscript fully available?

Reviewer #1: No

Reviewer #2: No

4. Is the manuscript presented in an intelligible fashion and written in standard English?

Reviewer #1: No

Reviewer #2: No

5. Review Comments to the Author

Reviewer #1: With respect to authors’ effort and considered subject in mining with potential high imposing damage in both human lives and project economy, this work doesn’t have any novelty and contribution. It overall follows a trivial routine procedure in geo-engineering practice to assess the hazard and risk and more like a technical draft for employer, where trivial evidence can be seen in section 2.2.

Anyway,

Literally:

- Poor quality English suffering from massive linguistic flaws and full of repeated words, long and vague statements. Simply can be seen from the first sentence in Abstract. Significant proofread by a native expert is mandatory.

- suffers from trivial improper headers and sub-headers. Study on the method!!!! lecturing on known theory??? Engineering application of what??? Research on mechanical parameters…???? calculation results????

- Superficial technical literature review. It doesn’t show any characterized problem statement, pursued goal, the gap of previous works, the limitation which is going to be filled here, the applied method and why, what motives for, the novelty, significant of contribution…. Required for more recent literature survey and comparison with them.

- Poor and dull Abstract. unreflective keywords (they should be representative and available in both Abstract and context. They should show the specificity of the work. What can be depicted from, for example numerical simulation? Partial displacement of what? equivalent parameter???? ….

- A series of ill-formatted references in the list.

- Unjustified conclusion. Almost unreadable, specifically #4. For example, ‘the potential landslide mode in the landslide area of the dump is the shear-bedding-bedding-bulging slide with the weak bed as the bottom interface’????

Technically:

The core of this work dealing with limit equilibrium method A) don’t consider the soil behavior, B) the precision of that due to several technical problem cannot be guaranteed. C) Referring to landslide movement, no consideration is given to the stress versus strain behavior of the soil.

Furthermore:

1) The map of the study area and corresponding hazard is opaque.

2) Just a total of 3 points are the used data!!!

3) in the case of numerical simulation, what are the used assumptions?? the number of mesh, type of mesh, the method of selection and reasons, definition of boundary conditions… are opaque.

3) How was the problem of SSI incorporated with the numerical model??

4) the effect of subsurface heterogeneity for spatial soil type/rock distribution in the study area is neglected. You are strongly recommended to have look at https://link.springer.com/article/10.1007/s10064-018-1400-9, https://link.springer.com/article/10.1007/s10706-016-9976-y, https://link.springer.com/article/10.1007/s10064-020-01922-8, https://www.sciencedirect.com/science/article/pii/S1674775521001165, https://www.mdpi.com/2220-9964/10/5/341 ...

5) Strongly would like to know the physical interpretation of Fig 4.

6) The FLAC just analyzes the nonlinearity of the model, not the actual structure and thus making different modeling assumptions will get different results for the same structure. Moreover, the achieved responses definitely are sensitive to the strengths and stiffnesses of the subsurface materials where based on #4 the actual properties may not be known accurately. Capacity design can greatly reduce uncertainty. No information regarding these problems and treatment methods have been given.

7) Each data type has some certain value of maximum and minimum, and if the data stored under certain datatype crosses these permissible values, then overflow and underflow occurs. This leads to truncation and rounds off errors in which your numerical model absolutely is incorporate with. So, how did you evaluate these errors and with what criteria?

9) Obvious poor discussion in terms of the accuracy performance, validation data, verification metrics, evidential analysis, and comparison with other models!!! Any discussion on limitations of presented method?? Can you prove the convergence or stability of the model??? Any explicit discussion to illustrate the limitations, pitfalls and practical difficulties of applied models under certainty??? How did you verify the fairness of made decisions while contextually it has different perspectives depending on the particular feed inputs? The evidence for generality??? The inclusion of the uncertainty involved in both the datasets and outputs???? Any reliability-based analysis? The effect of groundwater?? Looking at state-of the-art techniques like https://link.springer.com/article/10.1007/s11053-022-10051-w … can give you deep insights. Which one of the used inputs can more influence on the output? with what perspective you can calibrated the model? Looking at https://www.nature.com/articles/d41586-020-01812-9;
https://iwaponline.com/jh/article/22/3/562/72506/Updating-the-neural-network-sediment-load-models and https://www.sciencedirect.com/science/article/pii/S1364815218302822 … can help you.

10. Interpreting the evaluated results with the goals of this work??? Any limitations of the evaluation i.e., possible biases, validity/reliability of results???? Are there alternative explanations for your results???? How do your results compare with those of similar programs???? Have the different data collection methods used to measure your progress shown similar results??? Are your results consistent with theories supported by previous research??

11. Wondering with Table 1. Quaternary system??? How these parameters have been measured? For how many samples? In what spatial distribution? With what resolution? In which depth?

Reviewer #2: Authors have suggested a modified Morgenstern-Price method for stability analysis of mining dump. However, with out Figure illustrating the derivation of the method, it is very difficult to verify the correctness of the method. Hence, authors should provide a clear figure with layers and slices consideration for the derivation. Though, authors have written strips, but its not clear its vertical, inclined or horizontal strips. These types of method of slices can be considered depending upon the suitability. Its not clear why author's did not considered simplified Janbu's method , which is simpler.

However, as the authors presented their results with practical applications, authors are encouraged to submit the revised version with above description for adoptability of these techniques.

There are various syntax corrections to be made in the manuscript.

The writing style is very poor and needs to be written like a technical paper. Some specific comments are

The description of Figure 7 does not match with Figure illustration. There is no discussion about Figure 8 & 9.

There are other errors like Mohr-Coulomb is written like " Moll-Coulomb". The citation style "Chen Z Y et al.2005; Dong X

H et al.2012; Zhao T et al.2019" is also not proper.

Writing of Eq 5 is confusing, it should be in subscript style. There are similar various grammatical mistakes in the manuscript.

6. PLOS authors have the option to publish the peer review history of their article (what does this mean?). If published, this will include your full peer review and any attached files.

Reviewer #1: No

Reviewer #2: No

---

## [Author Response · Author response to Decision Letter 0]

14 Sep 2023

Dear Editors and Reviewers:

Thank you for your letter and for the reviewers’ comments concerning our manuscript entitled “Study on the control effect of local basement replacement on the stability of dump” (PONE-D-23-16609R1).” Those comments are all valuable and very helpful for revising and improving our paper, as well as the important guiding significance to our research. We have studied the comments carefully and have made corrections which we hope meet with approval. Revised portions are marked in red on the paper. The main corrections in the paper and the responses to the reviewer’s comments are as follows:

Respond to journal requests

Request 1. Please ensure that your manuscript meets PLOS ONE's style requirements, including those for file naming. The PLOS ONE style templates can be found at 

Response: The manuscript has been carefully revised in accordance with the stylistic requirements of PLOS ONE.

Request 2. Please note that PLOS ONE has specific guidelines on code sharing for submissions in which author-generated code underpins the findings in the manuscript. In these cases, all author-generated code must be made available without restrictions upon publication of the work. Please review our guidelines at https://journals.plos.org/plosone/s/materials-and-software-sharing#loc-sharing-code and ensure that your code is shared in a way that follows best practice and facilitates reproducibility and reuse.

Response: Our code has been shared according to your guidelines

Request 3. We note that you have provided funding information that is not currently declared in your Funding Statement. However, funding information should not appear in the Acknowledgments section or other areas of your manuscript. We will only publish funding information present in the Funding Statement section of the online submission form. 

 "This study was supported by The National Natural Science Foundation of China (52204135, 52204136), Liaoning Tens of Millions of Talents Program, Silk Road 1+1 Scientific Research Cooperation Project of Ministry of Education(P20210121076), Discipline Innovation Team of Liaoning Technical University (LNTU20TD-01) and International Science and Technology Cooperation Program of Liaoning Province(2022JH2/10700004)."

Response: I am very sorry that I wrote the funding statement in the manuscript. 

"This study was supported by The National Natural Science Foundation of China (52204135, 52204136), Liaoning Tens of Millions of Talents Program, Silk Road 1+1 Scientific Research Cooperation Project of Ministry of Education(P20210121076), Discipline Innovation Team of Liaoning Technical University (LNTU20TD-01) and International Science and Technology Cooperation Program of Liaoning Province(2022JH2/10700004)."

The above part is the whole funding project of the paper.

Request 4. Please state what role the funders took in the study. If the funders had no role, please state: "The funders had no role in study design, data collection and analysis, decision to publish, or preparation of the manuscript." 

Response: The funders had no role in study design, data collection and analysis, decision to publish, or preparation of the manuscript.

Request 5. In your Data Availability statement, you have not specified where the minimal data set underlying the results described in your manuscript can be found. PLOS defines a study's minimal data set as the underlying data used to reach the conclusions drawn in the manuscript and any additional data required to replicate the reported study findings in their entirety.

Response: All relevant data are within the manuscript and its Supporting Information files.

Request 6. We note that Figure 2 in your submission contain [map/satellite] images which may be copyrighted. All PLOS content is published under the Creative Commons Attribution License (CC BY 4.0), which means that the manuscript, images, and Supporting Information files will be freely available online, and any third party is permitted to access, download, copy, distribute, and use these materials in any way, even commercially, with proper attribution.

Response: The aerial photo in FIG. 3 is provided by the company that belongs to Jilin Guole open-cut Coal Mine in our engineering example in this paper. We have consulted the relevant copyright department of the company about the copyright issue of this picture and determined that FIG. 3 does not involve any copyright protection issue and is allowed to be used and distributed by third parties without restriction, even for commercial use and distribution.

Request 7. Please ensure that you refer to Figures 8 and 9 in your text as, if accepted, production will need this reference to link the reader to the figure.

Response: We have made the modification link in the revised manuscript, as shown in Figure 9,10.

Request 8. We note you have included a table to which you do not refer in the text of your manuscript. Please ensure that you refer to Table 3 in your text; if accepted, production will need this reference to link the reader to the Table.

Response: Sorry, this is our mistake, thank you for finding this problem, we have corrected it in the revised manuscript.

Responses to Reviewer 1

Question 1: With respect to authors’ effort and considered subject in mining with potential high imposing damage in both human lives and project economy, this work doesn’t have any novelty and contribution. It overall follows a trivial routine procedure in geo-engineering practice to assess the hazard and risk and more like a technical draft for employer, where trivial evidence can be seen in section 2.2.

Response: First of all, thank you very much for your careful reading of our team's manuscript. Judging from your point of view, this work does not have any novelty or contribution, but in practical engineering practice, there are indeed many technical problems like the ones encountered in this paper. In order to solve the contradiction of this project, the local filling technology of the base of the waste dump was studied by our team, With some successful cases, the stability of the slope of the mine dump is controlled.

Question 2: Poor quality English suffering from massive linguistic flaws and full of repeated words, long and vague statements. Simply can be seen from the first sentence in Abstract. Significant proofread by a native expert is mandatory.

Response: The poor quality of the English you speak and the flaws in the language have been carefully corrected in the revised manuscript.

Question 3: suffers from trivial improper headers and sub-headers. Study on the method!!!! lecturing on known theory??? Engineering application of what??? Research on mechanical parameters…???? calculation results????

Response: We have carefully revised the inappropriate title and subtitle you pointed out; The method study is an improvement of the Morgenstern-Price method. The shear strength parameters of soil coupled with basement replacement and filling are studied by using the soil dump of Guole No. 2 open-pit mine in Jilin Province as an engineering example.

Question 4: Superficial technical literature review. It doesn’t show any characterized problem statement, pursued goal, the gap of previous works, the limitation which is going to be filled here, the applied method and why, what motives for, the novelty, significant of contribution…. Required for more recent literature survey and comparison with them.

Response: The literature review section has been carefully revised based on the recent literature you recommended.

Question 5: Poor and dull Abstract. unreflective keywords (they should be representative and available in both Abstract and context. They should show the specificity of the work. What can be depicted from, for example numerical simulation? Partial displacement of what? equivalent parameter???? ….

Response: The summary section has been modified, for example “For practical ease, it is advisable to choose the lowest step in the dump's slope for construction convenience. As the local replacement rate of the base increases, the slope's stability coefficient gradually improves, with the K-Fs ratio showing a prominent role in this process.Additionally, numerical simulation methods are employed to elucidate the mechanism of the dump's landslide following local basement replacement, thereby providing comprehensive evidence of the engineering applicability of this method.The research results demonstrate a promising practical application prospect for effectively controlling the stability of soft base dump slopes.”

Question 6: A series of ill-formatted references in the list.

Response: Misformatted references in the list have been corrected.

Question 7: Unjustified conclusion. Almost unreadable, specifically #4. For example, ‘the potential landslide mode in the landslide area of the dump is the shear-bedding-bedding-bulging slide with the weak bed as the bottom interface’????

Response: In the fourth conclusion, through the mutual confirmation of numerical simulation results and two-dimensional limit equilibrium method, the potential landslide mode of the dump slope can be determined.

Question 8:Technically:The core of this work dealing with limit equilibrium method A) don’t consider the soil behavior, B) the precision of that due to several technical problem cannot be guaranteed. C) Referring to landslide movement, no consideration is given to the stress versus strain behavior of the soil.

Response:A)The change of mechanical properties of the soil coupled with the Tertiary clay after the excavation of the quaternary system is considered；B) ；C)To study the mechanical properties of the coupled soil by numerical simulation is to consider the stress-strain behavior of the coupled soil.

Question 9: The map of the study area and corresponding hazard is opaque.

Response:We replaced the aerial image with a DTM plane model and corrected the study area accordingly.

Question 10: Just a total of 3 points are the used data!!!

Response: The three points you mentioned have a small amount of data, so we have made serious consideration and increased the number of sample points to 6 to test the mechanical properties of coupled soil.

Question 11: in the case of numerical simulation, what are the used assumptions?? the number of mesh, type of mesh, the method of selection and reasons, definition of boundary conditions… are opaque.

Response: The assumptions used are: (1) In order to ensure the occurrence of the simulation process of the coupling soil property change, it is assumed that the natural clay of the quaternary system and the Tertiary system are completely fit and real soil after replacement, satisfying the continuity equation; (2) After the Tertiary clay is replaced, the local filling area becomes a water-retaining layer, assuming that the influence of the quaternary aquifer on the coupled soil is not considered; (3) The local replacement area of the base of the dump (the next step) can be approximately regarded as a uniform load.The number of grids is 87,747 cells and 89,619 nodes.Including the boundary and interface of the foundation pit, the soil filled with two kinds of Mohr-coulomb materials, and the initial boundary condition and loading mode are defined.

Question 12: How was the problem of SSI incorporated with the numerical model??

Response: After some assumptions, the numerical model is used to simulate the engineering case

Question 13: the effect of subsurface heterogeneity for spatial soil type/rock distribution in the study area is neglected. You are strongly recommended to have look at https://link.springer.com/article/10.1007/s10064-018-1400-9, https://link.springer.com/article/10.1007/s10706-016-9976-y, https://link.springer.com/article/10.1007/s10064-020-01922-8, https://www.sciencedirect.com/science/article/pii/S1674775521001165, https://www.mdpi.com/2220-9964/10/5/341 ...

Response: Thank you very much for the journal paper you recommended, which benefited me a lot. However, I think the numerical simulation study on mechanical parameters of soil replacement in the studied area in this paper takes into account the influence of the heterogeneity of rock and soil mass you mentioned.

Question 14: Strongly would like to know the physical interpretation of Fig 4.

Response: The shear strength parameters and deformation parameters of the coupled soil were obtained by the stress-strain curves of the coupled soil at different positions in the numerical simulation

Question 15: The FLAC just analyzes the nonlinearity of the model, not the actual structure and thus making different modeling assumptions will get different results for the same structure. Moreover, the achieved responses definitely are sensitive to the strengths and stiffnesses of the subsurface materials where based on #4 the actual properties may not be known accurately. Capacity design can greatly reduce uncertainty. No information regarding these problems and treatment methods have been given.

Response: As you said, it is difficult to know the actual mechanical parameters of the coupled soil after replacement through field tests, so this paper considers applying uniform load to the replacement area by numerical simulation, and then exploring the mechanical properties of the coupled soil.

Question 16: Each data type has some certain value of maximum and minimum, and if the data stored under certain datatype crosses these permissible values, then overflow and underflow occurs. This leads to truncation and rounds off errors in which your numerical model absolutely is incorporate with. So, how did you evaluate these errors and with what criteria?

Response:The curve data of stress and strain in the numerical model selected in this paper do not involve truncation and rounding errors you mentioned.

Question 17: Obvious poor discussion in terms of the accuracy performance, validation data, verification metrics, evidential analysis, and comparison with other models!!! Any discussion on limitations of presented method?? Can you prove the convergence or stability of the model??? Any explicit discussion to illustrate the limitations, pitfalls and practical difficulties of applied models under certainty??? How did you verify the fairness of made decisions while contextually it has different perspectives depending on the particular feed inputs? The evidence for generality??? The inclusion of the uncertainty involved in both the datasets and outputs???? Any reliability-based analysis? The effect of groundwater?? Looking at state-of the-art techniques like https://link.springer.com/article/10.1007/s11053-022-10051-w … can give you deep insights. Which one of the used inputs can more influence on the output? with what perspective you can calibrated the model? Looking at https://www.nature.com/articles/d41586-020-01812-9;
https://iwaponline.com/jh/article/22/3/562/72506/Updating-the-neural-network-sediment-load-models and https://www.sciencedirect.com/science/article/pii/S1364815218302822 … can help you.

Response: The improved Morgenstern-Price method proposed in this paper does have certain limitations, because the original rigid body limit equilibrium method is to conduct mechanical analysis from a two-dimensional perspective. The stability of the model you refer to is reflected in the three-position numerical simulation of the engineering example, which has a mutual verification process with the two-dimensional limit equilibrium method. The fairness of decision-making you mentioned is reflected in the on-site engineering practice. In the face of many restrictions on the mine, such as land expropriation, subsequent elevation and expansion of the waste dump, the only way with strong operability is to partially replace the base to improve the stability of the waste dump slope. As for the evidence of universality, I have supplemented it in the literature review. In addition, the influence of groundwater you mentioned has also been described in the paper. In many cases, the influence of groundwater on slope has been clearly reflected in the shear strength parameters of soil.

Question 18: Interpreting the evaluated results with the goals of this work??? Any limitations of the evaluation i.e., possible biases, validity/reliability of results???? Are there alternative explanations for your results???? How do your results compare with those of similar programs???? Have the different data collection methods used to measure your progress shown similar results??? Are your results consistent with theories supported by previous research??

Response: We believe that this is not in conflict. The purpose of this paper is to study how to improve the stability of the slope by replacing the base locally. Under the condition that the relevant criteria are met, this method has demonstrated engineering practicability. The engineering example is the application of the results obtained from the theoretical derivation, and the results are relatively reliable.

Question 19: Wondering with Table 1. Quaternary system??? How these parameters have been measured? For how many samples? In what spatial distribution? With what resolution? In which depth?

Response: In the process of analyzing the stability of slope, the shear strength and deformation parameters of soil mass should be used. Therefore, 5 parameter indexes are given in this paper. The relevant parameters are obtained by comprehensive means such as laboratory test and parameter reverse analysis, and have strong realistic reliability.

Responses to Reviewer 2

Question 1: Authors have suggested a modified Morgenstern-Price method for stability analysis of mining dump. However, with out Figure illustrating the derivation of the method, it is very difficult to verify the correctness of the method. Hence, authors should provide a clear figure with layers and slices consideration for the derivation. Though, authors have written strips, but its not clear its vertical, inclined or horizontal strips. These types of method of slices can be considered depending upon the suitability. Its not clear why author's did not considered simplified Janbu's method , which is simpler.

However, as the authors presented their results with practical applications, authors are encouraged to submit the revised version with above description for adoptability of these techniques.

Response: What you said is quite right. In the revised manuscript, I added the stress analysis diagram of the improved Morgenstern-Price method according to your valuable suggestions, and vertically divided the whole slope body slice. In addition, we have taken into account whether the simplified Janbu method mentioned by you should be considered. Although JANBU method is also based on the sliding surface of arbitrary morphology, this method is not easy to converge. Considering the engineering applicability, we chose Morgenstern-Price method to improve it.

Question 1:Some specific comments are

The description of Figure 7 does not match with Figure illustration. There is no discussion about Figure 8 & 9.

There are other errors like Mohr-Coulomb is written like " Moll-Coulomb". The citation style "Chen Z Y et al.2005; Dong X

H et al.2012; Zhao T et al.2019" is also not proper.

Writing of Eq 5 is confusing, it should be in subscript style. There are similar various grammatical mistakes in the manuscript.

Response: We have revised the errors in the articles you mentioned one by one, and the specific changes are in the revised manuscript.

---

## [Editor Report · Decision Letter 1]

2 Oct 2023

Study on the control effect of local basement replacement on the stability of dump

PONE-D-23-16609R1

Dear Dr. Li,

We’re pleased to inform you that your manuscript has been judged scientifically suitable for publication and will be formally accepted for publication once it meets all outstanding technical requirements.

Kind regards,

Sani Isah Abba, PhD

Academic Editor

PLOS ONE
---

## [Editor Report · Acceptance letter]

6 Oct 2023

PONE-D-23-16609R1 

Study on the control effect of local basement replacement on the stability of dump 

Dear Dr. Yin:

I'm pleased to inform you that your manuscript has been deemed suitable for publication in PLOS ONE. Congratulations! Your manuscript is now with our production department. 

Kind regards, 

on behalf of

Dr. Sani Isah Abba 

Academic Editor

PLOS ONE